# TAB-MIA: A BENCHMARK DATASET FOR MEMBERSHIP INFERENCE ATTACKS ON TABULAR DATA IN LLMS

**Eyal German, Sagiv Antebi, Daniel Samira, Asaf Shabtai, Yuval Elovici**
Faculty of Computer and Information Science
Ben-Gurion University of the Negev
Beer Sheva, Israel
{germane, sagivan, samirada}@post.bgu.ac.il
{shabtaia, elovici}@bgu.ac.il

## ABSTRACT

Large language models (LLMs) are increasingly trained on tabular data, which, unlike unstructured text, often contains personally identifiable information (PII) in a highly structured and explicit format. As a result, privacy risks arise, since sensitive records can be inadvertently retained by the model and exposed through data extraction or membership inference attacks (MIAs). While existing MIA methods primarily target textual content, their efficacy and threat implications may differ when applied to structured data, due to its limited content, diverse data types, unique value distributions, and column-level semantics. In this paper, we present Tab-MIA, a benchmark dataset for evaluating MIAs on tabular data in LLMs and demonstrate how it can be used. Tab-MIA comprises five data collections, each represented in six different encoding formats. Using our Tab-MIA benchmark, we conduct the first evaluation of state-of-the-art MIA methods on LLMs fine-tuned with tabular data across multiple encoding formats. In the evaluation, we analyze the memorization behavior of pretrained LLMs on structured data derived from Wikipedia tables. Our findings show that LLMs memorize tabular data in ways that vary across encoding formats, making them susceptible to extraction via MIAs. Even when fine-tuned for as few as three epochs, models exhibit high vulnerability, with AUROC scores approaching 90% in most cases. Tab-MIA enables systematic evaluation of these risks and provides a foundation for developing privacy-preserving methods for tabular data in LLMs.

## 1 INTRODUCTION

Large language models (LLMs) have emerged as core components of modern artificial intelligence (AI) systems due to their advanced language understanding and generation capabilities, supporting applications ranging from scientific discovery to natural, human-like interaction (Berti et al., 2025; Wei et al., 2022). These models are typically trained on vast and diverse datasets comprised of web content, academic publications, code repositories, and, increasingly, structured tabular data from organizational and public databases (Fang et al., 2024a; Paranjape et al., 2023).

Tabular data, such as financial spreadsheets and electronic health records, serve as the basis of data-driven workflows in healthcare, finance, public administration, and other sectors. Their structured format—rows as entities and columns as attributes—helps both humans and machine learning models learn patterns, relationships, and statistical properties efficiently. While LLMs have traditionally been developed and applied for unstructured textual data, recent research reflects the growing interest in adapting LLMs to effectively process such structured inputs by representing tables in text-like formats (Herzig et al., 2020; Yin et al., 2020; Narayan et al., 2022). This shift extends LLMs' capabilities to reasoning tasks involving both unstructured and structured data. However, incorporating tabular data in the training set of an LLM poses unique challenges and risks. Tabular data may contain personally identifiable information (PII), commercially sensitive material, or domain-specific details that are not intended for broad dissemination (Yeom et al., 2018a; Zeng et al., 2024).

LLMs, including those trained on structured data, can memorize and leak sensitive records since they are vulnerable to *membership inference attacks* (MIAs), in which an adversary attempts to determine whether a particular record was included in the model's training set (Shokri et al., 2017a; Carlini et al., 2022a). These attacks typically rely on subtle differences in the model's behavior when queried with examples it has seen during training compared to unseen examples (Cao et al., 2023; Hu et al., 2022).

MIAs on LLMs have been studied extensively in the context of textual data, where researchers typically analyze confidence scores at the sentence- or paragraph-level to detect training set membership (Song et al., 2025; Duan et al., 2024). These studies generally assume that the models were trained on free-form, unstructured text—such as natural language sentences and documents. Tabular data, which is often heterogeneous, may exhibit skewed value distributions and contain explicit column-level semantics, making both the design of MIAs and the development of effective defenses more challenging (Borisov et al., 2022a; Fang et al., 2024a).

Recent work has shown that generative models can effectively interpret, transform, and synthesize tabular data (Zha et al., 2023), and other studies have shown that the choice of table encoding format—such as JSON, HTML, Markdown, or Key-Value Pair—can impact model performance (Fang et al., 2024a). However, the studies primarily focused on improving task accuracy and generalization, with comparatively little research attention given to understanding memorization risks or the potential exploitation of tabular data through MIAs. Prior research has shown that LLM performance is highly sensitive to the input format: for instance, DFLoader and JSON have been found effective for fact-finding and transformation tasks (Singha et al., 2023), while HTML and XML outperform plain-text formats like CSV or X-separated values in table QA and field-value prediction (Sui et al., 2023; 2024a). This performance gap is often attributed to the prevalence of web-based markup (e.g., HTML) in the pretraining data of models like GPT-3.5 and GPT-4 (OpenAI, 2024b), making them more effective at processing tables serialized in familiar, structured input styles.

In this paper, we present Tab-MIA, a benchmark dataset specifically designed to evaluate MIAs against LLMs fine-tuned on tabular data. Tab-MIA includes five collections consisting of tables, each represented in six different textual encoding formats. To our knowledge, this is the first comprehensive evaluation of MIAs on LLMs trained with structured tabular data across multiple encoding formats. We systematically examine the sensitivity of LLMs to MIAs under various conditions, including after fine-tuning with a limited number of epochs on tabular datasets, and in the pretrained setting, where the pretrained model is assumed to be trained on a tabular subset of Wikipedia. In our experiments, various configurations of models, data encodings, and training epochs are examined.

One evaluation shows that LLMs can memorize tabular data to a degree sufficient for effective membership inference. Notably, even when fine-tuned for as few as three epochs, attack success rates can be high, with AUROC scores approaching 90%. We also observed partial transferability of attacks across encoding formats, indicating that adversaries may succeed without exact knowledge of the specific format used in training. These findings highlight the need for privacy-preserving training practices when training LLMs on structured data. Our work broadens the scope of MIA research, which has largely not focused on structured data, and highlights the need for privacy-preserving strategies designed to address the challenges posed by the unique characteristics of tabular formats.

The main contributions of our paper are (1) we present the first benchmark dataset to evaluate MIAs against LLMs trained on tabular data; (2) we conduct the first evaluation of state-of-the-art (SOTA) MIAs on LLMs fine-tuned with tabular data across multiple encoding formats; and (3) we analyze the memorization behavior of recent SOTA LLMs on structured data derived from Wikipedia tables.

## 2  RELATED WORK

LLMs have demonstrated promising capabilities in handling structured data across tasks such as tabular representation, question answering, and data generation. In this section, we review prior work focused on: (1) MIAs on LLMs, (2) encoding-strategy-based methods for using tabular data with LLMs, and (3) emerging risks when incorporating structured data into LLM training sets.

## 2.1 MEMBERSHIP INFERENCE ATTACKS ON LLMS

MIAs (Shokri et al., 2017b) aim to determine whether a given sample $x$ is part of a training set $D_{\text{train}}$ of a model $f$. An attacker receives a sample $x$ and the trained model $f$, and applies an attack model $A$ to classify $x$ as a member $A(f(x)) = 1$, or non-member otherwise. MIAs against LLMs have received increasing attention (Carlini et al., 2022b; Mattern et al., 2023; Zhang et al., 2024). Recent studies categorized MIA methods into *reference-based* and *reference-free* approaches (Antebi et al., 2025). Reference-based attacks primarily rely on training shadow models to mimic the target model's behavior. A prominent example is LiRA (Carlini et al., 2022b), which estimates the likelihood ratio of a sample's loss under two model output distributions, one where the sample was included in training and one where it was not. In addition, Carlini et al. (Carlini et al., 2023) show that memorization in LLMs scales predictably with model size, data duplication, and context length, providing strong empirical evidence of extractable training data and highlighting why reference-based MIAs can succeed. While often effective, such methods are computationally expensive, as they require training multiple shadow models and calibrating their outputs.

Reference-free attacks rely on confidence metrics derived from a single model's output. The *LOSS attack (PPL)* (Yeom et al., 2018b) infers membership based on the model's loss value relative to a fixed threshold. The *Zlib attack* (Carlini et al., 2021) uses the ratio of log-likelihood to its Zlib compression length, while the *Neighbor attack* (Mattern et al., 2023) examines perplexity shifts by substituting words with similar tokens generated by an auxiliary model. More recently, *Min-K%* (Shi et al., 2024) and *Min-K%++* (Zhang et al., 2025) were shown to improve attack efficiency by averaging the lowest probability tokens, with Min-K%++ further applying normalization over log probabilities. In addition, the authors of *RECALL* (Xie et al., 2024), *DC-PDD* (Zhang et al., 2024), and *Tag&Tab* (Antebi et al., 2025) introduced more advanced strategies that improve MIA performance on LLMs compared to other methods.

## 2.2 LLMS AND TABULAR DATA

Many enterprise and scientific datasets consist of tabular data, which is composed of rows and columns of structured attributes (Fang et al., 2024b). Traditional tree-based models such as XG-Boost (Chen & Guestrin, 2016) and LightGBM (Ke et al., 2017) have long been dominant for tabular data tasks, particularly due to their effectiveness on small-to-medium sized datasets and strong inductive biases for numerical features (Gorishniy et al., 2021). However, recent research has explored the use of LLMs for tabular data applications, including classification, regression, data augmentation, data generation, and table-based QA (Sui et al., 2024a; Borisov et al., 2022b; Ding et al., 2023). LLMs use their strengths, such as in-context generalization and instruction following, to better understand serialized tables, handle numeric or categorical features, and produce flexible outputs, even in scenarios that conventional machine learning models struggle with. LLMs support table-based tasks such as *Table QA*, *fact verification*, and *Text2SQL* (Chen, 2023a; Ye et al., 2023). Earlier methods like TAPAS (Herzig et al., 2020) and TaBERT (Yin et al., 2020) used specialized encoders, while modern LLMs process table queries by serializing them as text or leveraging external code calls (Sui et al., 2024a; Liu et al., 2023).

A central challenge in applying LLMs to tabular data lies in how to represent structured tables in a text-based input format suitable for transformer architectures. Prior work proposed serializing tables using various strategies, including natural language templates, JSON, Markdown, HTML, and Key-Value Pair (Dinh et al., 2022; Slack & Singh, 2023; Jaitly et al., 2023). The choice of serialization affects not only model performance but also how well the structure and semantics of the table are preserved. For example, Hegselmann et al. (2023) proposed TabLLM, a method that systematically evaluates multiple table encoding formats. Their evaluation showed that simple natural language patterns, such as "The [column] is [value]," can yield strong performance across a range of tabular classification tasks, likely due to their alignment with the model's pretraining distributions.

Although LLMs can process moderately sized serialized tables, handling very large tables remains challenging due to the transformers' fixed-length context window. This restricts the amount of tabular data a model can process in a single input, making it difficult to handle large tables without partitioning or truncation (Sui et al., 2024a;b), which can disrupt the model's ability to capture long-range dependencies and global relationships across rows and columns. To address this, compression-based frameworks like SHEETENCODER (Dong et al., 2024) have been developed. SHEETENCODER

reduces the size of table inputs by selecting structural anchors, applying inverted-index translation to remove redundancy, and aggregating similar numeric fields, thereby preserving important relational information while remaining within context window limits.

While prior research optimized table serialization for accuracy and scalability, it largely overlooked the privacy implications of different serialization strategies. Tab-MIA fills this gap by systematically evaluating how encoding choices affect memorization and membership inference risk.

## 2.3 Privacy Risks When Training LLMs with Structured Data

Integrating structured tabular data in LLMs offers substantial benefits for data-driven reasoning, enabling models to combine natural language understanding with structured data processing (Gorishniy et al., 2021; Fang et al., 2024b). However, it also introduces distinct privacy and security risks that differ from those encountered when training on unstructured text. A critical vulnerability stems from the fact that tabular datasets often contain sensitive information, such as personal identifiers, financial records, or medical details, that are highly susceptible to memorization (Carlini et al., 2022b; Lukas et al., 2023). Even seemingly benign fields, when combined, can form distinctive patterns that compromise individuals' privacy. Once such information is memorized by a model, it may be vulnerable to extraction via MIAs, exposing individual records or sensitive attributes (Carlini et al., 2021).

While MIAs have been widely studied in the context of unstructured text corpora, such as books, Wikipedia, and web documents (Xie et al., 2024; Antebi et al., 2025), there is a notable lack of benchmark datasets for structured tabular data. Existing MIA benchmark datasets like BookMIA, WikiMIA (Shi et al., 2024), and MIMIR (Duan et al., 2024) have helped characterize MIA risks in textual domains, but they do not consider the unique structural format that is present in tabular datasets. The MIDST benchmark (Membership Inference over Diffusion-models-based Synthetic Tabular Data) (Organizers, 2025) extends this landscape by evaluating MIAs on diffusion models trained to synthesize tabular data. However, MIDST focuses on privacy risks in synthetic data generation, where sensitive records may be reconstructed from the denoising trajectory. In contrast, our Tab-MIA benchmark addresses a different privacy risk: memorization of tabular records in LLMs fine-tuned on serialized tables, where leakage occurs through token-level probabilities tied to column semantics. This distinction highlights complementary attack surfaces. To address the LLM-specific risk, our Tab-MIA benchmark evaluates membership inference on tabular datasets across diverse encoding formats and LLM configurations.

## 3 Construction of the Tab-MIA Benchmark

Our goal in constructing the Tab-MIA benchmark is to facilitate the systematic evaluation of how MIAs can be applied to extract the tabular data used to fine-tune LLMs. Unlike text-based benchmarks, which focus on sentences or paragraphs, tabular benchmarks must handle heterogeneous types of columns, various encoding formats, and repeated patterns across structurally similar tables. By creating a controlled yet realistic set of tables from publicly available datasets, Tab-MIA enables systematic evaluation of how different table-encoding strategies affect vulnerability to MIAs. We use it to analyze how different formats affect memorization and attack performance.

### 3.1 Datasets

The benchmark integrates real-world datasets widely used in language modeling and tabular machine learning, covering diverse structural characteristics and application domains. To enable systematic evaluation of MIA risks in LLMs fine-tuned using tabular data, Tab-MIA includes datasets representing both **short-context** and **long-context** tables.

Short-context tables are derived from QA benchmarks in which each instance originally pairs a question with a supporting table. In our setting, we discard the question text and retain only the *unique tables* to focus on tabular memorization effects. We include WikiTableQuestions (WTQ) (Pasupat & Liang, 2015), WikiSQL (Zhong et al., 2017), and TabFact (Chen et al., 2020). Long-context tables are derived from structured tabular benchmarks frequently used in fairness, regression, and privacy studies. We include the Adult (Census Income) dataset (Becker & Kohavi, 1996) and the California

```
(a) JSON

[
  {"Name": "Alice", "Age": 30},
  {"Name": "Bob",   "Age": 25},
  {"Name": "Carol", "Age": 28}
]

(c) Markdown

| Name  | Age |
|-------|-----|
| Alice | 30  |
| Bob   | 25  |
| Carol | 28  |

(e) Key-is-Value

Name is Alice. Age is 30.
Name is Bob.   Age is 25.
Name is Carol. Age is 28.
```

```
(b) HTML

<table>
  <tr><th>Name</th><th>Age</th></tr>
  <tr><td>Alice</td><td>30</td></tr>
  <tr><td>Bob</td><td>25</td></tr>
  <tr><td>Carol</td><td>28</td></tr>
</table>

(d) Key-Value Pair

Name: Alice | Age: 30
Name: Bob   | Age: 25
Name: Carol | Age: 28

(f) Line-Separated

Name,Age
Alice,30
Bob,25
Carol,28
```

Figure 1: The same $3 \times 2$ table snippet serialized into the six encoding formats used in the Tab-MIA benchmark: (a) JSON, (b) HTML, (c) Markdown, (d) Key-Value Pair, (e) Key-is-Value, and (f) Line-Separated (CSV-like).

Housing dataset (Pace & Barry, 1997). Due to input length limitations inherent to LLMs, long tables are segmented into row-wise *chunks* sized to fit within the model's context window while preserving structural coherence. A full summary of the datasets used in Tab-MIA, including record counts before and after filtering, feature dimensionality, context type (short or long), and data sources, is provided in Table 1.

Table 1: Summary of datasets used in Tab-MIA.

| Name | Short/Long | # Records | # After Filter | # Features | Based On |
|---|---|---|---|---|---|
| WTQ | Short | 2,108 | 1,290 | $\geq 5$ | Wikipedia |
| WikiSQL | Short | 24,241 | 17,900 | $\geq 5$ | Wikipedia |
| TabFact | Short | 16,573 | 13,100 | $\geq 5$ | Wikipedia |
| Adult (Census Income) | Long | 48,842 | 2,440 | 15 | US Census |
| California Housing | Long | 20,640 | 1,030 | 10 | US Housing Survey |

## 3.2 DATA PREPARATION

To construct the Tab-MIA benchmark, we processed each of its constituent datasets using a standardized pipeline designed to ensure data quality, consistency, and experimental control. First, we perform a filtering and deduplication step to ensure that each table appears only once in the benchmark, preventing artificial inflation of the memorization signal due to repeated exposure. Next, we apply context-specific processing to match the model's input length constraints. For *short-context tables*, we filter out any table whose serialized representation in the Line-Separated format exceeds 10,000 characters, removing overly large tables that could dominate training dynamics or introduce truncation artifacts. To accommodate *long-context tables*, we split each table into chunks of 20 records each to fit within the model's input length constraints and maintain consistency across samples.

Each resulting table (or table chunk, in the case of long-context tables) is serialized into multiple textual formats to investigate how the encoding style influences memorization. We use six encoding strategies, each reflecting a different structural abstraction of the table (illustrated in Figure 1).

All encoded variants are saved as JSONL files to support reproducible experiments. Encoding each table in multiple ways enables us to systematically examine whether certain formats result in greater memorization by the model, and whether some styles are inherently more resistant to MIAs.

## 4 EXPERIMENTAL SETUP

We evaluate the vulnerability of fine-tuned LLMs to MIA under various configurations of models, data encodings, and training epochs. We fine-tune four SOTA open-weight language models—LLaMA-3.1 8B, LLaMA-3.2 3B (Meta Team, 2024), Gemma-3 4B (Gemma Team, 2025), and Mistral 7B (Jiang et al., 2023)—which have diverse training objectives, tokenizer variants, and parameter scales. All models are trained using QLoRA (Dettmers et al., 2023), a parameter-efficient fine-tuning (PEFT) method leveraging 4-bit quantized weights. Unless otherwise specified, models are fine-tuned for three epochs; however, in our analysis of training length, we also explore the effect of varying the number of epochs between one and three. In each training run, half of the tables are used as member records while the remainder serve as non-members. Additional details on the hyperparameters are provided in Appendix A.1.

To assess the privacy risk, we consider three black-box MIAs: the *LOSS attack (PPL)* (Yeom et al., 2018b), which relies on negative log-likelihood scores; the *Min-K% attack* (Shi et al., 2024), which averages the lowest $k\%$ token probabilities to identify memorized content; and *Min-K%++ attack* (Zhang et al., 2025), which normalizes log probabilities before aggregation to examine robustness to length and calibration effects. For each attack, we report two standard metrics, AUROC and TPR@FPR=5% (Carlini et al., 2022b), measuring detection performance across decision thresholds and under strict privacy constraints, respectively.

We analyze the empirical findings of our experiments using Tab-MIA, to address four key questions. First, to what extent does the choice of table encoding format impact LLMs' memorization and vulnerability to MIAs? Second, how does varying the number of fine-tuning epochs affect LLMs' vulnerability to MIAs? Third, to what extent do MIAs remain effective when the encoding format used during detection differs from the format used during model fine-tuning? Fourth, to what extent do publicly available *pretrained* LLMs memorize public tabular data?

## 5 RESULTS

In this section, we present our empirical findings using the Tab-MIA benchmark to evaluate MIAs on tabular data in LLMs. The results highlight consistent trends in vulnerability driven by fine-tuning duration, encoding format, and model architecture.

### 5.1 EFFECT OF ENCODING FORMAT

Table 2: Comparison of the AUROC scores achieved by different MIA methods across table encoding formats and models on the *California Housing* dataset. Bold values indicate the highest score per row (encoding), while underlined values indicate the highest score per column (model-method pair).

| Encoding Method | Llama-3.2 3B | | | Mistral 7B | | | Gemma-3 4B | | |
|---|---|---|---|---|---|---|---|---|---|
| | PPL | Min-K 20.0% | Min-K++ 20.0% | PPL | Min-K 20.0% | Min-K++ 20.0% | PPL | Min-K 20.0% | Min-K++ 20.0% |
| Markdown | 60.60 | 60.90 | 72.00 | 65.60 | 73.10 | **80.00** | 59.10 | 64.10 | 67.80 |
| JSON | 59.60 | 59.60 | 53.00 | **61.40** | **61.40** | 54.50 | 58.40 | 58.40 | 55.00 |
| HTML | 59.70 | 59.70 | 55.80 | **61.70** | **61.70** | 50.60 | 59.10 | 61.20 | 55.40 |
| Key-Value Pair | 62.80 | 62.80 | 78.70 | 72.40 | 74.70 | **92.60** | 59.30 | 60.80 | 67.00 |
| Key-is-Value | 60.20 | 60.20 | 55.10 | 63.70 | 65.00 | **74.90** | 59.20 | 60.60 | 66.70 |
| Line-Separated | 61.60 | 64.90 | 77.20 | 69.70 | 84.90 | **86.80** | 62.30 | 72.10 | 73.80 |

Table 3: Comparison of the AUROC scores achieved by different MIA methods across table encoding formats and models on the *WTQ* dataset. Bold values indicate the highest score per row (encoding), while underlined values indicate the highest score per column (model-method pair).

| Encoding Method | Llama-3.2 3B | | | Mistral 7B | | | Gemma-3 4B | | |
|---|---|---|---|---|---|---|---|---|---|
| | PPL | Min-K 20.0% | Min-K++ 20.0% | PPL | Min-K 20.0% | Min-K++ 20.0% | PPL | Min-K 20.0% | Min-K++ 20.0% |
| Markdown | 68.00 | 69.50 | 85.30 | 87.00 | 88.40 | **94.20** | 73.70 | 74.80 | 86.70 |
| JSON | 67.10 | 67.50 | 79.80 | 79.40 | 79.60 | **82.70** | 70.70 | 71.00 | 79.20 |
| HTML | 66.30 | 66.60 | 79.70 | 82.80 | 83.00 | **92.90** | 72.10 | 72.80 | 83.30 |
| Key-Value Pair | 67.00 | 67.80 | 83.50 | 85.00 | 85.70 | **94.90** | 72.80 | 73.80 | 85.50 |
| Key-is-Value | 67.00 | 67.90 | 83.70 | 83.60 | 84.20 | **89.70** | 72.30 | 73.20 | 85.00 |
| Line-Separated | 70.40 | 72.40 | 89.70 | 87.30 | 90.40 | **97.70** | 74.70 | 76.50 | 89.60 |

Textual encoding shapes the way tabular structures are presented to LLMs and can influence their tendency to memorize data. In this experiment, we fine-tuned the models and executed the MIAs on the datasets, using different encoding formats to assess their impact on the privacy risk. Tables 2 and 3 present the AUROC scores for MIAs on the *California Housing* (long-context) and *WTQ* (short-context) datasets, using the six examined encoding formats. On both datasets, the *Line-Separated* and *Key-Value Pair* formats exhibit the greatest vulnerability to membership inference. On the *WTQ* dataset, an AUROC of 97.7% with Mistral 7B was obtained using the *Line-Separated* format, and on the *California Housing* dataset, an AUROC of 92.6% was achieved using the *Key-Value Pair* format. These findings show that encoding format impacts the privacy risk. Flat, row-based encodings like Line-Separated and Key-Value Pair produce long, continuous sequences of content tokens that align closely with tokenizer boundaries. This structure concentrates learning on individual cell values, increasing the likelihood of memorization—resulting in the highest AUROC scores across datasets and MIA methods. In contrast, formats such as HTML and JSON introduce structural redundancy via tags and punctuation. This disperses model attention across non-content tokens, leading to lower AUROC scores—typically 10 points lower—indicating reduced memorization. Intermediate formats like Key-is-Value and Markdown strike a balance between structural clarity and redundancy, yielding moderate vulnerability. These results align with theoretical analyses showing that memorization risk increases with the effective input context length (Carlini et al., 2022b; 2023). Additional results are available in Appendix A.2.

## 5.2 EFFECT OF THE NUMBER OF FINE-TUNING EPOCHS

Table 4: AUROC scores for the *Min-K++ 20.0%* MIA on each dataset, evaluated on tables encoded in the Line-Separated format, as a function of the number of fine-tuning epochs. Bold values highlight the best-performing dataset per row.

| Model | # Epochs | Adult | California | WTQ | WikiSQL | TabFact |
|---|---|---|---|---|---|---|
| LLaMA-3.1 8B | 1 | 55.10 | 59.00 | 61.60 | 64.50 | **64.90** |
| | 2 | 60.00 | 72.80 | **80.80** | 78.60 | 79.60 |
| | 3 | 71.10 | 87.80 | **93.60** | 88.90 | 89.90 |
| Llama-3.2 3B | 1 | 54.10 | 57.70 | 57.60 | **61.50** | 61.50 |
| | 2 | 58.00 | 66.80 | **74.80** | 73.60 | 73.40 |
| | 3 | 64.40 | 77.20 | **89.70** | 83.20 | 80.40 |
| Mistral 7B | 1 | 54.60 | 57.80 | **69.70** | 67.50 | 68.50 |
| | 2 | 58.90 | 70.30 | **88.40** | 80.00 | 81.20 |
| | 3 | 71.50 | 86.80 | **97.70** | 87.80 | 89.90 |
| Gemma-3 4B | 1 | 53.90 | 54.30 | 59.30 | 62.60 | **63.30** |
| | 2 | 58.90 | 62.50 | 77.00 | 76.60 | **77.90** |
| | 3 | 67.70 | 73.80 | **89.60** | 86.10 | 87.40 |

Table 5: MIA results on the *WikiSQL* dataset for all examined models fine-tuned for 1, 2, and 3 epochs. Tables are encoded in the Line-Separated format. Bold values highlight the best-performing method per row.

| Model | # Epochs | PPL | | Min-K 20.0% | | Min-K++ 20.0% | |
|---|---|---|---|---|---|---|---|
| | | AUROC | TPR@FPR=5% | AUROC | TPR@FPR=5% | AUROC | TPR@FPR=5% |
| Llama-3.2 3B | 1 | 55.90 | 7.40 | 56.40 | 7.60 | **61.50** | **7.90** |
| | 2 | 62.50 | 10.80 | 63.70 | 11.10 | **73.60** | **14.40** |
| | 3 | 69.40 | 15.90 | 71.20 | 16.10 | **83.20** | **25.30** |
| LLaMA-3.1 8B | 1 | 58.10 | 8.70 | 58.60 | 8.60 | **64.50** | **10.60** |
| | 2 | 67.20 | 15.20 | 68.40 | 15.30 | **78.60** | **22.80** |
| | 3 | 76.50 | 25.30 | 78.10 | 25.90 | **88.90** | **40.20** |
| Mistral 7B | 1 | 60.10 | 9.60 | 61.30 | 9.90 | **67.50** | **14.10** |
| | 2 | 68.40 | 15.40 | 70.50 | 16.60 | **80.00** | **26.20** |
| | 3 | 75.10 | 22.20 | 77.60 | 23.80 | **87.80** | **42.90** |
| Gemma-3 4B | 1 | 56.20 | 7.40 | 56.60 | 7.60 | **62.60** | **8.70** |
| | 2 | 64.00 | 11.30 | 64.90 | 11.70 | **76.60** | **18.60** |
| | 3 | 72.50 | 17.60 | 73.90 | 18.70 | **86.10** | **34.30** |

MIAs generally rely on the assumption that models are expected to exhibit greater memorization of training data as the number of fine-tuning epochs increases. This motivates examining how the number of fine-tuning epochs impacts privacy leakage for various models and attack methods. To this end, we fine-tuned each model for 1, 2, and 3 epochs on the tabular datasets included in our benchmark and evaluated the MIAs' success. For this experiment, the tables were serialized into the Line-Separated encoding format.

Table 4 presents the results for the *Min-K++ 20.0%* MIA for each of the datasets. We observe a consistent and substantial increase in vulnerability as the number of fine-tuning epochs grows. This trend holds across all models and datasets. The effect is especially pronounced in short-context datasets, particularly on the *WTQ* dataset, where AUROC scores reach as high as 97.7% with Mistral 7B after three epochs and exceed 89.6% across all models. In contrast, long-context datasets exhibit more moderate vulnerability. For example, on the *Adult* dataset, the highest AUROC is 71.5% with Mistral 7B, and on *California Housing*, the highest result is 87.8% with LLaMA-3.1 8B. Table 5, which compares the performance of the examined attacks on the *WikiSQL* dataset, illustrates the trends discussed above in greater detail. For all attacks, as fine-tuning progresses, vulnerability increases, with higher AUROC scores obtained as the number of epochs grew across models. Among them, *Min-K++ 20.0%* consistently performs the best, achieving an AUROC of 88.9 with LLaMA-3.1 8B and 87.8 with Mistral 7B. Additional results for the remaining datasets and attack methods are provided in Appendix A.3.

MIAs generally achieve higher AUROC scores against larger models such as `LLaMA-3.1 8B` and `Mistral 7B`, compared to smaller models like `LLaMA-3.2 3B` and `Gemma-3 4B`. For example, after fine-tuning for three epochs, with tables encoded using the Line-Separated format on the California Housing dataset, the *Min-K++ 20.0%* MIA achieves AUROC scores of 86.8% and 87.8% respectively with `Mistral 7B` and `LLaMA-3.1 8B`, compared to 77.2% and 73.8% with `LLaMA-3.2 3B` and `Gemma-3 4B`. Chen (2023b) found that larger models offer clear advantages in table reasoning tasks, highlighting the performance benefits of increased scale. However, our results reveal a corresponding privacy trade-off: larger models are also significantly more vulnerable to MIAs, with differences of nearly 10 to 14 percentage points in AUROC compared to smaller LLMs. While prior work attributes such susceptibility to the greater memorization capacity of LLMs (Carlini et al., 2023; 2021), our findings extend this observation to models fine-tuned on tabular data, where increased model size correlates with greater leakage under MIAs.

## 5.3 CROSS-FORMAT GENERALIZATION

In this experiment, we examine whether tabular data learned during fine-tuning with one table encoding format remains detectable by MIAs applied using a different format. This scenario mirrors real-world deployment settings, where the encoding format used during the model's training is unknown. To evaluate this, we fine-tuned the *Gemma-3 4B* model on the *TabFact* dataset using one of the six encoding formats and executed the *Min-K++ 20.0%* attack. The results, shown in Figure 2, reveal partial cross-format generalization: memorization signals often persist even when the evaluation format differs from the training format. Diagonal cells (where training and evaluation formats match) tend to yield the highest AUROC values, confirming that MIAs are most effective when structural representations align. For example, training and evaluating on the Markdown format yields an AUROC of 85.2%, whereas switching the attack format to Key-Value or Line-Separated reduces performance to 68.9% and 69.4%, respectively.

This aligns with prior findings by Kandpal et al. (Kandpal et al., 2022), who show that memorization in LLMs is highly sensitive to the exact structure and representation of the training data: even small deviations from the training representation—such as changes in format—can substantially reduce the detectability of memorized content. Our results similarly show that misalignment between training and detection encodings weakens MIA performance, but importantly does not eliminate it. In several cases, strong memorization signals persist across formats, indicating that LLMs can still leak training information even when the attacker does not know the original encoding. This suggests that cross-format generalization remains a meaningful privacy risk.

To gain additional insights, we compute the average AUROC values across the rows and columns of the heatmap. These averages reflect how effective each encoding format is when used to encode the data for MIA detection (rows) and for model fine-tuning (columns). The most vulnerable format for

MIA detection is HTML (76.0), followed by Key-Value Pair (73.2) and JSON (71.2), suggesting that these formats offer greater advantages to attackers. On the training side, Line-Separated and Key-is-Value induce the most memorization, resulting in average AUROCs of 74.6 and 72.8, respectively. From a defender's perspective, selecting training formats like JSON or HTML, which yield lower average AUROCs of 69.4 and 70.1, may help reduce privacy risk.

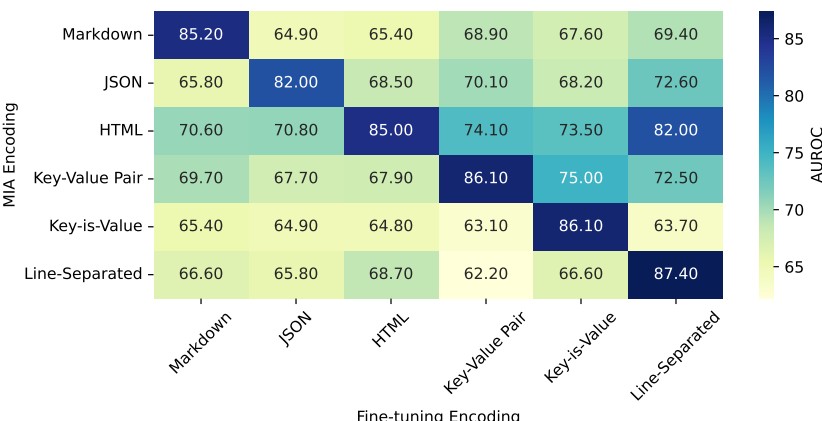

Figure 2: Heatmap showing the AUROC achieved by the *Min-K++ 20.0%* MIA on the *WTQ* dataset using the *Gemma-3 4B* model. Each cell compares the encoding used during fine-tuning (columns) with the encoding used during MIA detection (rows).

## 5.4 PRETRAINED MODELS

Table 6: AUROC scores achieved by the *Min-K++ 20.0%* MIA on the *WTQ* dataset using pretrained models without fine-tuning. Synthetic data was generated to serve as non-member samples. The table compares performance across table encoding formats for each model. Bold values indicate the highest score per row (encoding), while underlined values indicate the highest score per column (model).

| Encoding Method | Llama-3.1 8B | Llama-3.2 3B | Mistral 7B | Gemma-3 4B |
|---|---|---|---|---|
| Markdown | **69.30** | 62.20 | 63.00 | 60.70 |
| JSON | **62.40** | 57.60 | 59.90 | 58.40 |
| HTML | **66.70** | 60.00 | 61.70 | 61.80 |
| Key-Value Pair | **72.00** | 66.20 | 66.90 | 63.40 |
| Key-is-Value | **71.60** | 65.90 | 64.10 | 61.90 |
| Line-Separated | **71.50** | 63.80 | 62.90 | 60.90 |

In this experiment, we assess LLMs' vulnerability to MIAs in their pretrained state—prior to any fine-tuning. Our goal is to determine whether publicly available models have inadvertently memorized examples from the WTQ dataset, which forms part of our benchmark. Given WTQ's wide use and its reliance on Wikipedia tables, we assume that its contents may have been included in the pretraining corpora of many open-weight LLMs. To simulate an MIA setting, we treated the original WTQ tables as member samples and generated synthetic non-member tables using the *GPT-4o mini* (OpenAI, 2024a) model (see Appendix A.4 for details on the generation process). We then used the MIN-K++ 20.0% attack to test each pretrained model for evidence of memorization of the WTQ tables. Table 6 presents the AUROC scores for four models with the six encoding formats. The results show pretrained models without further fine-tuning exhibit moderate levels of data leakage. The highest AUROC of 72.0 is observed for LLaMA-3.1 8B with the Key-Value Pair format. Formats such as Key-Value Pair, Key-is-Value, and Line-Separated consistently result in greater vulnerability across models, with AUROC scores frequently exceeding 60%, indicating that the models likely memorized these tables during pretraining.

## 6 CONCLUSION

Tab-MIA is the first benchmark for evaluating MIAs on LLMs trained on tabular data. Through controlled experiments on four SOTA open-source LLMs and six encoding strategies, our experiments show that fine-tuning LLMs on tabular data might cause memorization and thus make them vulnerable to MIAs. Some attacks can achieve AUROC scores exceeding 95% with minimal fine-tuning, underscoring the risk of memorization and privacy leakage. In contrast, we find that using encodings that introduce syntactic noise (e.g., verbose or structured formats such as HTML or JSON) mitigates attack success. Our benchmark provides a foundation for the systematic evaluation of privacy risks in various scenarios with different models and table encoding formats.

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

## A  TECHNICAL APPENDICES AND SUPPLEMENTARY MATERIAL

### A.1  TRAINING AND EVALUATION CONFIGURATIONS

This appendix contains the training configurations used in our experiments. All models are fine-tuned using QLoRA (Dettmers et al., 2023), a PEFT method that enables efficient training with 4-bit quantized weights. Fine-tuning is performed using a batch size of two on a single `RTX 6000` GPU (48GB VRAM). We apply a learning rate of `3e-4`, use the `paged_adamw_8bit` optimizer, and set `warmup_steps` to 20. We use a fixed random seed of 42 for all dataset splits and data loading to ensure reproducibility.

For each dataset, 50% of the tables are selected as member records for fine-tuning, with the remaining used as non-members for MIA evaluation. All experiments are implemented using HuggingFace Transformers and PEFT libraries, with evaluation scripts provided in the public code repository.

### A.2  IMPACT OF TABLE ENCODING FORMATS ON MIA PERFORMANCE

This section provides detailed results on the effect of different table encoding formats on models' susceptibility to MIAs. Tables 7- 11 report the AUROC and TPR@FPR=5% values for six encoding schemes (HTML, JSON, Key-Value Pair, Key-is-Value, Line-Separated, and Markdown) for all model–dataset configurations.

### A.3  EFFECT OF FINE-TUNING EPOCHS ON MIA VULNERABILITY

This section presents comprehensive results on how the number of fine-tuning epochs affects vulnerability to MIAs across all model–dataset configurations in our benchmark. For this experiment, we report results using the Line-Separated encoding format, as it consistently exhibits high memorization rates across models and datasets, making it a strong representative for analyzing privacy risk. Tables 12–15 summarize AUROC and TPR@FPR=5% metrics across three representative MIA methods: LOSS (PPL), Min-K 20.0%, and Min-K++ 20.0%. Across all methods, we observe that longer fine-tuning leads to increased model memorization and thus greater vulnerability to MIAs.

Table 7: MIA results on the *Adult* dataset for the examined models, with the various encoding formats. For each method, both AUROC and TPR@FPR=5% are reported. Bold values highlight the best-performing method per row.

| Model | Encoding | PPL | | Min-K 20.0% | | Min-K++ 20.0% | |
|---|---|---|---|---|---|---|---|
| | | AUROC | TPR@FPR=5% | AUROC | TPR@FPR=5% | AUROC | TPR@FPR=5% |
| LLaMA-3.1 8B | HTML | **62.40** | **9.30** | 62.40 | 9.20 | 54.90 | 6.50 |
| | JSON | **69.90** | **17.30** | 69.90 | 17.30 | 68.80 | 14.50 |
| | Key-is-Value | **70.50** | **18.30** | 70.40 | 17.80 | 70.50 | 15.40 |
| | Key-Value Pair | **72.60** | 21.90 | 72.60 | **22.00** | 71.30 | 16.50 |
| | Line-Separated | 73.90 | 24.20 | **74.30** | **25.80** | 71.10 | 15.50 |
| | Markdown | **75.70** | 27.60 | 75.70 | **28.20** | 73.20 | 19.70 |
| Llama-3.2 3B | HTML | 61.60 | **9.70** | 61.60 | 9.70 | **62.70** | 8.80 |
| | JSON | 61.40 | 9.30 | 61.40 | 9.30 | **63.70** | **9.40** |
| | Key-is-Value | 60.50 | 8.80 | 60.40 | 8.70 | **63.10** | **9.50** |
| | Key-Value Pair | 60.20 | 8.10 | 60.20 | 8.40 | **63.00** | **9.40** |
| | Line-Separated | 62.40 | 9.60 | 62.60 | 9.80 | **64.40** | **10.20** |
| | Markdown | 62.80 | 9.80 | 62.80 | 9.80 | **65.10** | **10.90** |
| Mistral 7B | HTML | 71.00 | 17.70 | 71.00 | 17.60 | **75.30** | **21.90** |
| | JSON | **56.90** | 5.80 | 56.90 | **5.90** | 50.90 | 3.80 |
| | Key-is-Value | 67.40 | 12.50 | 67.40 | 12.60 | **73.30** | **19.40** |
| | Key-Value Pair | 66.90 | 10.90 | 67.00 | 11.00 | **72.40** | **15.50** |
| | Line-Separated | 65.90 | 10.80 | 67.40 | 11.80 | **71.50** | **14.70** |
| | Markdown | 71.60 | 14.40 | 71.90 | 14.70 | **78.20** | **27.30** |
| Gemma-3 4B | HTML | **59.20** | 7.90 | 59.20 | **8.20** | 54.20 | 7.60 |
| | JSON | **55.70** | 6.80 | 55.70 | 6.70 | 50.80 | **8.00** |
| | Key-is-Value | 57.40 | 6.50 | 57.40 | 6.60 | **59.80** | 6.80 |
| | Key-Value Pair | 57.60 | 7.00 | 57.60 | **7.10** | **59.80** | 6.80 |
| | Line-Separated | 63.00 | 10.50 | 64.80 | **12.00** | **67.70** | 11.70 |
| | Markdown | 58.20 | 7.10 | 58.40 | 7.00 | **61.30** | **8.00** |

Table 8: MIA results on the *California Housing* dataset for the examined models, with the various encoding formats. For each method, both AUROC and TPR@FPR=5% are reported. Bold values highlight the best-performing method per row.

| Model | Encoding | PPL | | Min-K 20.0% | | Min-K++ 20.0% | |
|---|---|---|---|---|---|---|---|
| | | AUROC | TPR@FPR=5% | AUROC | TPR@FPR=5% | AUROC | TPR@FPR=5% |
| LLaMA-3.1 8B | HTML | 69.40 | 21.30 | 69.40 | 21.10 | **88.90** | **56.80** |
| | JSON | **63.80** | **15.10** | 63.80 | 15.10 | 54.60 | 8.30 |
| | Key-is-Value | **64.30** | **14.30** | 64.30 | 14.10 | 56.00 | 9.50 |
| | Key-Value Pair | 68.30 | 18.40 | 68.20 | 18.40 | **88.20** | **51.20** |
| | Line-Separated | 66.30 | 19.60 | 70.40 | 19.60 | **87.80** | **52.50** |
| | Markdown | 64.60 | 15.50 | 64.90 | 15.90 | **80.00** | **34.50** |
| Llama-3.2 3B | HTML | **59.70** | **13.20** | 59.70 | 13.00 | 55.80 | 7.00 |
| | JSON | **59.60** | 12.20 | 59.60 | **12.40** | 53.00 | 4.50 |
| | Key-is-Value | **60.20** | **13.60** | 60.20 | 13.60 | 55.10 | 10.10 |
| | Key-Value Pair | 62.80 | 16.10 | 62.80 | 15.90 | **78.70** | **26.20** |
| | Line-Separated | 61.60 | 14.90 | 64.90 | 14.00 | **77.20** | **26.60** |
| | Markdown | 60.60 | 13.20 | 60.90 | 13.00 | **72.00** | **22.10** |
| Mistral 7B | HTML | **61.70** | **13.40** | 61.70 | 13.40 | 50.60 | 5.00 |
| | JSON | **61.40** | **14.10** | 61.40 | 14.00 | 54.50 | 6.80 |
| | Key-is-Value | 63.70 | 15.50 | 65.00 | 14.70 | **74.90** | **28.70** |
| | Key-Value Pair | 72.40 | 24.20 | 74.70 | 24.20 | **92.60** | **68.00** |
| | Line-Separated | 69.70 | 19.40 | 84.90 | 45.00 | **86.80** | **56.80** |
| | Markdown | 65.60 | 17.60 | 73.10 | 23.10 | **80.00** | **39.10** |
| Gemma-3 4B | HTML | 59.10 | 11.00 | **61.20** | **11.80** | 55.40 | 7.80 |
| | JSON | **58.40** | **11.40** | 58.40 | 11.40 | 55.00 | 7.60 |
| | Key-is-Value | 59.20 | 10.30 | 60.60 | 11.00 | **66.70** | **15.70** |
| | Key-Value Pair | 59.30 | 11.20 | 60.80 | 11.20 | **67.00** | **15.30** |
| | Line-Separated | 62.30 | 13.20 | 72.10 | 20.70 | **73.80** | **23.30** |
| | Markdown | 59.10 | 11.20 | 64.10 | 13.20 | **67.80** | **15.30** |

## A.4 SYNTHETIC TABLES GENERATION

In Section 5.4, we evaluate pretrained LLMs for evidence of memorization of public tabular datasets. To simulate a MIA setting in this scenario, we required *non-member* tables that resemble the structure of the WTQ dataset but do not duplicate any of its records. For this purpose, we generated synthetic tables using a controlled LLM-based synthesis procedure.

Table 9: MIA results on the *WTQ* dataset for the examined models, with the various encoding formats. For each method, both AUROC and TPR@FPR=5% are reported. Bold values highlight the best-performing method per row.

| Model | Encoding | PPL | | Min-K 20.0% | | Min-K++ 20.0% | |
|---|---|---|---|---|---|---|---|
| | | AUROC | TPR@FPR=5% | AUROC | TPR@FPR=5% | AUROC | TPR@FPR=5% |
| LLaMA-3.1 8B | HTML | 71.60 | 17.30 | 71.80 | 17.30 | **82.00** | **41.70** |
| | JSON | 72.90 | 19.10 | 73.20 | 19.30 | **81.70** | **44.60** |
| | Key-is-Value | 73.20 | 21.20 | 73.90 | 21.20 | **86.50** | **50.20** |
| | Key-Value Pair | 73.80 | 23.20 | 74.40 | 22.90 | **86.70** | **51.80** |
| | Line-Separated | 77.90 | 27.20 | 79.50 | 29.50 | **93.60** | **66.40** |
| | Markdown | 74.40 | 19.80 | 75.60 | 20.50 | **89.40** | **54.70** |
| Llama-3.2 3B | HTML | 66.30 | 10.40 | 66.60 | 10.60 | **79.70** | **28.60** |
| | JSON | 67.10 | 11.70 | 67.50 | 11.70 | **79.80** | **33.00** |
| | Key-is-Value | 67.00 | 12.80 | 67.90 | 13.10 | **83.70** | **38.30** |
| | Key-Value Pair | 67.00 | 12.10 | 67.80 | 12.80 | **83.50** | **40.40** |
| | Line-Separated | 70.40 | 14.80 | 72.40 | 16.30 | **89.70** | **48.40** |
| | Markdown | 68.00 | 12.60 | 69.50 | 13.50 | **85.30** | **31.90** |
| Mistral 7B | HTML | 82.80 | 29.70 | 83.00 | 30.00 | **92.90** | **70.00** |
| | JSON | 79.40 | 29.20 | 79.60 | 29.10 | **82.70** | **53.80** |
| | Key-is-Value | 83.60 | 34.70 | 84.20 | 35.60 | **89.70** | **68.10** |
| | Key-Value Pair | 85.00 | 37.60 | 85.70 | 38.60 | **94.90** | **79.20** |
| | Line-Separated | 87.30 | 47.00 | 90.40 | 51.30 | **97.70** | **88.20** |
| | Markdown | 87.00 | 36.70 | 88.40 | 36.90 | **94.20** | **84.00** |
| Gemma-3 4B | HTML | 72.10 | 12.90 | 72.80 | 12.90 | **83.30** | **42.30** |
| | JSON | 70.70 | 12.10 | 71.00 | 12.10 | **79.20** | **37.00** |
| | Key-is-Value | 72.30 | 14.50 | 73.20 | 14.60 | **85.00** | **46.50** |
| | Key-Value Pair | 72.80 | 15.60 | 73.80 | 15.70 | **85.50** | **49.30** |
| | Line-Separated | 74.70 | 16.50 | 76.50 | 20.20 | **89.60** | **54.10** |
| | Markdown | 73.70 | 16.30 | 74.80 | 16.80 | **86.70** | **49.10** |

Table 10: MIA results on the *WikiSQL* dataset for the examined models, with the various encoding formats. For each method, both AUROC and TPR@FPR=5% are reported. Bold values highlight the best-performing method per row.

| Model | Encoding | PPL | | Min-K 20.0% | | Min-K++ 20.0% | |
|---|---|---|---|---|---|---|---|
| | | AUROC | TPR@FPR=5% | AUROC | TPR@FPR=5% | AUROC | TPR@FPR=5% |
| LLaMA-3.1 8B | HTML | 74.50 | 18.90 | 74.50 | 18.90 | **81.90** | **30.40** |
| | JSON | 75.20 | 20.10 | 75.30 | 20.10 | **82.70** | **32.40** |
| | Key-is-Value | 75.50 | 20.90 | 75.80 | 21.10 | **86.10** | **37.60** |
| | Key-Value Pair | 75.60 | 21.20 | 75.80 | 21.40 | **86.00** | **36.30** |
| | Line-Separated | 76.50 | 25.30 | 78.10 | 25.90 | **88.90** | **40.20** |
| | Markdown | 75.60 | 20.20 | 76.10 | 20.70 | **86.20** | **33.70** |
| Llama-3.2 3B | HTML | 67.90 | 11.70 | 68.00 | 11.80 | **76.80** | **19.50** |
| | JSON | 69.10 | 13.10 | 69.20 | 13.10 | **78.50** | **22.40** |
| | Key-is-Value | 64.60 | 8.70 | 65.10 | 8.80 | **72.30** | **12.60** |
| | Key-Value Pair | 69.10 | 13.40 | 69.50 | 13.50 | **80.70** | **23.10** |
| | Line-Separated | 69.40 | 15.90 | 71.20 | 16.10 | **83.20** | **25.30** |
| | Markdown | 68.10 | 11.50 | 69.00 | 11.70 | **71.40** | **14.60** |
| Mistral 7B | HTML | 72.20 | 16.50 | 72.30 | 16.50 | **79.60** | **31.00** |
| | JSON | 72.90 | 16.80 | 73.10 | 16.70 | **80.30** | **30.70** |
| | Key-is-Value | 74.70 | 19.50 | 75.40 | 19.80 | **85.30** | **37.50** |
| | Key-Value Pair | 74.60 | 19.90 | 75.20 | 20.20 | **85.40** | **37.70** |
| | Line-Separated | 75.10 | 22.20 | 77.60 | 23.80 | **87.80** | **42.90** |
| | Markdown | 75.10 | 19.80 | 76.20 | 20.40 | **86.10** | **39.10** |
| Gemma-3 4B | HTML | 72.00 | 16.00 | 72.50 | 16.20 | **83.40** | **29.20** |
| | JSON | 71.20 | 14.40 | 71.30 | 14.40 | **80.00** | **26.90** |
| | Key-is-Value | 72.00 | 15.70 | 72.50 | 15.80 | **84.20** | **30.30** |
| | Key-Value Pair | 71.90 | 15.80 | 72.50 | 15.80 | **84.50** | **31.20** |
| | Line-Separated | 72.50 | 17.60 | 73.90 | 18.70 | **86.10** | **34.30** |
| | Markdown | 71.20 | 14.30 | 71.90 | 14.60 | **83.00** | **26.80** |

We implemented a Python pipeline that reads the original tabular data and queries the `GPT-4o-mini` model to produce synthetic replacements for each table. The pipeline preserves the table's schema and formatting, but replaces cell values with realistic, non-identical alternatives. This ensures that synthetic records maintain semantic plausibility while preventing verbatim overlap with the original dataset.

Table 11: MIA results on the *TabFact* dataset for the examined models, with the various encoding formats. For each method, both AUROC and TPR@FPR=5% are reported. Bold values highlight the best-performing method per row.

| Model | Encoding | PPL | | Min-K 20.0% | | Min-K++ 20.0% | |
|---|---|---|---|---|---|---|---|
| | | AUROC | TPR@FPR=5% | AUROC | TPR@FPR=5% | AUROC | TPR@FPR=5% |
| LLaMA-3.1 8B | HTML | 76.10 | 19.50 | 76.10 | 19.50 | **83.70** | **34.50** |
| | JSON | 70.30 | 13.00 | **70.40** | 13.00 | 69.40 | **21.20** |
| | Key-is-Value | 77.60 | 21.70 | 77.90 | 21.90 | **88.30** | **41.10** |
| | Key-Value Pair | 78.10 | 22.10 | 78.30 | 22.40 | **88.40** | **42.10** |
| | Line-Separated | 77.40 | 26.40 | 78.70 | 27.00 | **89.90** | **47.00** |
| | Markdown | 78.00 | 22.20 | 78.70 | 22.80 | **88.70** | **40.50** |
| Llama-3.2 3B | HTML | 68.60 | 11.40 | 68.70 | 11.40 | **78.20** | **20.30** |
| | JSON | 69.50 | 11.90 | 69.60 | 12.00 | **80.20** | **23.30** |
| | Key-is-Value | 64.10 | 8.60 | 64.70 | 8.80 | **71.90** | **12.20** |
| | Key-Value Pair | 67.70 | 11.60 | 68.20 | 11.60 | **78.80** | **20.60** |
| | Line-Separated | 67.80 | 13.80 | 69.40 | 14.00 | **80.40** | **31.10** |
| | Markdown | 67.70 | 10.90 | 68.80 | 11.20 | **73.90** | **12.80** |
| Mistral 7B | HTML | 74.60 | 18.60 | 74.60 | 18.60 | **82.40** | **37.60** |
| | JSON | 75.70 | 19.60 | 75.80 | 19.60 | **83.90** | **39.10** |
| | Key-is-Value | 76.80 | 21.00 | 77.40 | 21.10 | **87.70** | **43.30** |
| | Key-Value Pair | 76.90 | 21.80 | 77.50 | 21.90 | **88.00** | **43.80** |
| | Line-Separated | 77.00 | 24.60 | 78.90 | 25.50 | **89.90** | **50.50** |
| | Markdown | 77.50 | 23.10 | 78.70 | 23.60 | **89.10** | **47.20** |
| Gemma-3 4B | HTML | 72.40 | 15.50 | 72.90 | 15.60 | **85.00** | **33.10** |
| | JSON | 72.00 | 14.20 | 72.20 | 14.20 | **82.00** | **30.90** |
| | Key-is-Value | 72.70 | 14.90 | 73.10 | 15.10 | **86.10** | **33.00** |
| | Key-Value Pair | 72.50 | 14.70 | 72.90 | 14.80 | **86.10** | **33.90** |
| | Line-Separated | 72.70 | 17.40 | 73.60 | 18.20 | **87.40** | **37.40** |
| | Markdown | 71.80 | 14.90 | 72.50 | 15.20 | **85.20** | **29.60** |

Table 12: MIA results on the *Adult* dataset for all examined models fine-tuned for 1, 2, and 3 epochs. Tables are encoded in the Line-Separated format. Bold values highlight the best-performing method per row.

| Model | # Epochs | PPL | | Min-K 20.0% | | Min-K++ 20.0% | |
|---|---|---|---|---|---|---|---|
| | | AUROC | TPR@FPR=5% | AUROC | TPR@FPR=5% | AUROC | TPR@FPR=5% |
| Llama-3.2 3B | 1 | 53.30 | 5.20 | 53.30 | 4.90 | **54.10** | **5.30** |
| | 2 | 56.50 | 7.00 | 56.50 | 6.30 | **58.00** | **7.30** |
| | 3 | 62.40 | 9.60 | 62.60 | 9.80 | **64.40** | **10.20** |
| LLaMA-3.1 8B | 1 | 53.80 | 6.30 | 53.70 | 6.40 | **55.10** | **6.70** |
| | 2 | 58.10 | 7.50 | 58.10 | **8.40** | **60.00** | 8.00 |
| | 3 | 73.90 | 24.20 | **74.30** | **25.80** | 71.10 | 15.50 |
| Mistral 7B | 1 | 54.00 | **5.20** | 54.10 | 4.40 | **54.60** | 5.20 |
| | 2 | 57.10 | **6.80** | 57.60 | 6.10 | **58.90** | 6.80 |
| | 3 | 65.90 | 10.80 | 67.40 | 11.80 | **71.50** | **14.70** |
| Gemma-3 4B | 1 | 53.20 | **6.20** | 53.10 | 5.70 | **53.90** | 5.00 |
| | 2 | 56.70 | 7.30 | 57.20 | 6.10 | **58.90** | **7.50** |
| | 3 | 63.00 | 10.50 | 64.80 | **12.00** | **67.70** | 11.70 |

**Prompt Used for Synthesis.** The following prompt was provided to the model for each table:

```
You are a data synthesizer. Your task is to generate a synthetic version
of the given tabular dataset for use in membership inference attack
evaluation on tabular data.
- Preserve the original table's structure, column names, and formatting.
- Change the values so that they are realistic but not identical to the original
  data.
- Output *only* the synthetic table|no explanations, no preamble,
  and no additional text.

Input:
The original table:
{table}
```

Table 13: MIA results on the *California Housing* dataset for all examined models fine-tuned for 1, 2, and 3 epochs. Tables are encoded in the Line-Separated format. Bold values highlight the best-performing method per row.

| Model | # Epochs | PPL | | Min-K 20.0% | | Min-K++ 20.0% | |
|---|---|---|---|---|---|---|---|
| | | AUROC | TPR@FPR=5% | AUROC | TPR@FPR=5% | AUROC | TPR@FPR=5% |
| Llama-3.2 3B | 1 | 53.90 | **8.30** | 55.20 | 7.40 | **57.70** | 7.40 |
| | 2 | 57.00 | 12.00 | 59.00 | 9.90 | **66.80** | **15.50** |
| | 3 | 61.60 | 14.90 | 64.90 | 14.00 | **77.20** | **26.60** |
| LLaMA-3.1 8B | 1 | 54.10 | 9.30 | 55.30 | 8.50 | **59.00** | **10.70** |
| | 2 | 58.70 | 13.40 | 61.10 | 11.20 | **72.80** | **22.70** |
| | 3 | 66.30 | 19.60 | 70.40 | 19.60 | **87.80** | **52.50** |
| Mistral 7B | 1 | 55.00 | 9.50 | 57.30 | 10.10 | **57.80** | **12.40** |
| | 2 | 59.70 | 13.40 | 68.20 | 18.80 | **70.30** | **23.60** |
| | 3 | 69.70 | 19.40 | 84.90 | 45.00 | **86.80** | **56.80** |
| Gemma-3 4B | 1 | 53.80 | **9.70** | **54.30** | 9.30 | 54.30 | 7.90 |
| | 2 | 56.90 | 10.70 | 61.40 | **14.10** | 62.50 | 12.60 |
| | 3 | 62.30 | 13.20 | 72.10 | 20.70 | **73.80** | **23.30** |

Table 14: MIA results on the *WTQ* dataset for all examined models fine-tuned for 1, 2, and 3 epochs. Tables are encoded in the Line-Separated format. Bold values highlight the best-performing method per row.

| Model | # Epochs | PPL | | Min-K 20.0% | | Min-K++ 20.0% | |
|---|---|---|---|---|---|---|---|
| | | AUROC | TPR@FPR=5% | AUROC | TPR@FPR=5% | AUROC | TPR@FPR=5% |
| Llama-3.2 3B | 1 | 51.50 | 3.70 | 51.90 | 5.10 | **57.60** | **5.90** |
| | 2 | 59.70 | 8.20 | 60.80 | 8.70 | **74.80** | **19.00** |
| | 3 | 70.40 | 14.80 | 72.40 | 16.30 | **89.70** | **48.40** |
| LLaMA-3.1 8B | 1 | 53.70 | 5.10 | 54.10 | 5.10 | **61.60** | **9.00** |
| | 2 | 64.70 | 10.70 | 65.80 | 12.00 | **80.80** | **30.50** |
| | 3 | 77.90 | 27.20 | 79.50 | 29.50 | **93.60** | **66.40** |
| Mistral 7B | 1 | 58.40 | 8.60 | 59.80 | 7.80 | **69.70** | **15.40** |
| | 2 | 74.30 | 20.80 | 76.80 | 21.20 | **88.40** | **55.20** |
| | 3 | 87.30 | 47.00 | 90.40 | 51.30 | **97.70** | **88.20** |
| Gemma-3 4B | 1 | 52.50 | 4.20 | 53.00 | 3.70 | **59.30** | **7.50** |
| | 2 | 61.90 | 9.50 | 62.90 | 9.00 | **77.00** | **24.90** |
| | 3 | 74.70 | 16.50 | 76.50 | 20.20 | **89.60** | **54.10** |

```
Output:
The synthetic table:
```

This process was only applied in the pretrained-model evaluation, where synthetic non-members were paired with WTQ member tables. For all fine-tuned experiments described in Section 4, non-member samples were drawn directly from the benchmark datasets without synthesis.

## A.5 DISCLOSURE OF LLM USAGE

In accordance with the ICLR 2026 policy on large language model (LLM) usage, we disclose that LLMs were used solely to aid and polish the writing of this manuscript. Their role was limited to improving grammar, clarity, and readability. No part of the research design, data processing, experimental implementation, analysis, or conclusions relied on LLM-generated content. All scientific contributions were conceived and executed entirely by the authors.

Table 15: MIA results on the *TabFact* dataset for all examined models fine-tuned for 1, 2, and 3 epochs. Tables are encoded in the Line-Separated format. Bold values highlight the best-performing method per row.

| Model | # Epochs | PPL | | Min-K 20.0% | | Min-K++ 20.0% | |
|---|---|---|---|---|---|---|---|
| | | AUROC | TPR@FPR=5% | AUROC | TPR@FPR=5% | AUROC | TPR@FPR=5% |
| Llama-3.2 3B | 1 | 55.10 | 6.60 | 55.50 | 6.50 | **61.50** | **8.80** |
| | 2 | 62.10 | 9.90 | 63.10 | 10.10 | **73.40** | **14.90** |
| | 3 | 67.80 | 13.80 | 69.40 | 14.00 | **80.40** | **31.10** |
| LLaMA-3.1 8B | 1 | 57.90 | 7.90 | 58.30 | 8.20 | **64.90** | **11.20** |
| | 2 | 67.80 | 15.10 | 68.90 | 15.20 | **79.60** | **24.20** |
| | 3 | 77.40 | 26.40 | 78.70 | 27.00 | **89.90** | **47.00** |
| Mistral 7B | 1 | 60.00 | 9.60 | 60.70 | 10.00 | **68.50** | **14.70** |
| | 2 | 69.20 | 16.00 | 70.60 | 16.70 | **81.20** | **29.40** |
| | 3 | 77.00 | 24.60 | 78.90 | 25.50 | **89.90** | **50.50** |
| Gemma-3 4B | 1 | 55.40 | 7.00 | 55.40 | 7.00 | **63.30** | **10.80** |
| | 2 | 63.80 | 11.40 | 64.40 | 11.70 | **77.90** | **20.10** |
| | 3 | 72.70 | 17.40 | 73.60 | 18.20 | **87.40** | **37.40** |

