# OpenReview forum: "Tab-MIA: A Benchmark Dataset for Membership Inference Attacks on Tabular Data in LLMs"
_ICLR.cc/2026/Conference — ICLR 2026 Poster_

### Official Review · Reviewer_bFS1 · 2025-10-22

**Soundness:** 3
**Presentation:** 4
**Contribution:** 3
**Rating:** 6
**Confidence:** 4

**Summary:**

The paper introduces a benchmark for membership inference attacks on tabular data in LLM finetuning.

Authors argue that many prior works have focused on evaluating MIAs on unstructured text included in LLM training data, while increasingly LLMs are also trained on more structured, tabular data. At the same time, data entries in tabular data might also contain sensitive information, and the way they are encoded might lead to different trends in privacy risk.

In this work, they propose a benchmark to evaluate MIAs on tabular data encoded and included in LLM finetuning. They consider 6 encoding formats, 5 datasets, 3 MIAs, 4 models and analyze the results. They confirm that larger models, trained for more epochs memorize more, and find that encodings using less redundancies are more at risk. They also find that their attacks generalize across formats, and apply their MIAs to LLM pretraining.

**Strengths:**

- The idea of specifically looking at privacy risk associated with structured/tabular data in LLM training data is, from what I know of the literature, novel and interesting.
- The paper is well written and clearly structured.
- I find the related work very well written and easy to follow. It might be worth adding this recent work of reference-based MIAs against LLMs [1].
- The work has an extensive evaluation framework, considering 6 different encoding formats, multiple models (and sizes), different regimes of memoriation by varying the number of epochs, 3 different MIAs and 5 datasets.
- Section 5.3 on cross-format generalization is a nice addition. You might want to link this to what others have found to be a 'mosaic memory', i.e. LLMs are very good at piecing together near-duplicate information, even when random tokens separate the information meaningful for membership [2].

[1] Hayes, J., Shumailov, I., Choquette-Choo, C. A., Jagielski, M., Kaissis, G., Lee, K., ... & Cooper, A. F. (2025). Strong Membership Inference Attacks on Massive Datasets and (Moderately) Large Language Models. arXiv preprint arXiv:2505.18773.

[2] Shilov, I., Meeus, M., & de Montjoye, Y. A. (2024). The Mosaic Memory of Large Language Models. arXiv preprint arXiv:2405.15523.

**Weaknesses:**

When discussing the results, I find that, at least in section 5.1, authors devote a substantial amount of time on memorization trends that are already quite well-known and in are in my opinion not the most interesting ones in this benchmark. For instance, the fact that AUC increases with model size or with number of epochs is already well known [1, 2, 3]. Moreover, the absolute value of AUC is ultimately a factor of the finetuning parameters such as learning rate, and is not really what matters in this case. Instead, the relative performance across tabular datasets, and encoding formats is more interesting and worth exploring a bit deeper in my opinion.

For instance, the difference in MIA performance across datasets in Table 2 is quite striking and interesting. Could you explain this by e.g. examining the length of the records (as longer sequences are more at risk [1,2,3]) or by examining how unique the records are (either in perplexity using the base model as in [3] or in overlap between members and non-members as explored in [4], or simply by analyzing the uniqueness of attributes across records).

The intuition why certain encoding formats are more at risk than others is quite interesting. However, I find the argument a bit confusing at the moment, i.e. you first mention that methods like HTML and JSON introduce structural redundancies so lower AUC, while in lines 405-406 you mention that longer context would increase the AUC. How are both arguments compatible? It might also be interesting to look at how efficiently each encoding format is tokenized by the target model.

- Results in Section 5.4 might be questionable. Namely, prior work has found that if members and non-members do not come from exactly the same distribution, any MIA performance might be due to the attack being able to detect a distribution shift rather than the target model memorizing the members [4, 5, 6, 7]. To understand this, I think it's important to include a blind/model-less baseline as in [6,7]. If this baseline is substantially better than a random guess, it would be hard to interpret any of the results from Table 6.

[1] Kandpal, N., Wallace, E., & Raffel, C. (2022, June). Deduplicating training data mitigates privacy risks in language models. In International Conference on Machine Learning (pp. 10697-10707). PMLR.

[2] Carlini, N., Ippolito, D., Jagielski, M., Lee, K., Tramer, F., & Zhang, C. (2022, February). Quantifying memorization across neural language models. In The Eleventh International Conference on Learning Representations.

[3] Meeus, M., Shilov, I., Faysse, M., & de Montjoye, Y. A. Copyright Traps for Large Language Models. In Forty-first International Conference on Machine Learning.

[4] Duan, M., Suri, A., Mireshghallah, N., Min, S., Shi, W., Zettlemoyer, L., ... & Hajishirzi, H. (2024). Do membership inference attacks work on large language models?. arXiv preprint arXiv:2402.07841.

[5] Maini, P., Jia, H., Papernot, N., & Dziedzic, A. (2024). LLM Dataset Inference: Did you train on my dataset?. Advances in Neural Information Processing Systems, 37, 124069-124092.

[6] Meeus, M., Shilov, I., Jain, S., Faysse, M., Rei, M., & de Montjoye, Y. A. (2025, April). Sok: Membership inference attacks on llms are rushing nowhere (and how to fix it). In 2025 IEEE Conference on Secure and Trustworthy Machine Learning (SaTML) (pp. 385-401). IEEE.

[7 ]Das, D., Zhang, J., & Tramèr, F. (2025, May). Blind baselines beat membership inference attacks for foundation models. In 2025 IEEE Security and Privacy Workshops (SPW) (pp. 118-125). IEEE.

**I hope these suggestions help improve the paper and, if addressed, I'd be willing to increase my score.**

**Questions:**

- Is there evidence that people have been finetuning or pretraining LLMs on tabular dataset including PII? It could be a nice addition to provide an overview of this.

- Could authors also add the Ref or Ratio attack from [1] or as used by [2]? You could easily just divide the target finetuned model loss by the loss of the pretrained model. From my experience, this should be better than any of the other methods you currently have.


[1] Carlini, N., Tramer, F., Wallace, E., Jagielski, M., Herbert-Voss, A., Lee, K., ... & Raffel, C. (2021). Extracting training data from large language models. In 30th USENIX security symposium (USENIX Security 21) (pp. 2633-2650).

[2] Duan, M., Suri, A., Mireshghallah, N., Min, S., Shi, W., Zettlemoyer, L., ... & Hajishirzi, H. (2024). Do membership inference attacks work on large language models?. arXiv preprint arXiv:2402.07841.

---

> ### Author Response · Authors · 2025-11-15
> **Response to Reviewer**
>
> Thank you for the positive assessment and detailed, technically grounded suggestions. We respond point-by-point. We will also update the related work and results sections as we suggested.
>
> **W1:** Section 5.1 focuses on known trends; deeper dataset-level analysis desirable
>
> **A1:** Agreed. We will shorten Sec. 5.1 and expand the dataset-level analysis (row uniqueness, row length, entropy), as these factors strongly correlate with the AUROC differences observed across datasets. We will also reorder the results sections so that the more informative analyses—such as encoding effects and dataset-level characteristics—appear earlier and receive clearer emphasis in the narrative.
>
> **W2:** Confusion: structural redundancy lowers AUC vs. longer context increasing AUC
>
> **A2:** We will clarify:
>
> Semantic length (content tokens) increases memorization.
> Syntactic redundancy (markup, tags) reduces memorization.
> These operate in opposite directions, explaining the observed patterns.
>
> **W3:** Pretrained results might reflect distribution shift; need blind baseline
>
> **A3:** Thank you for raising this important point. In the pretrained setting, member and non-member samples may indeed originate from slightly different distributions, since we cannot control which Wikipedia tables were included in the model’s pretraining corpus. To reduce this risk, we use synthetic non-member tables that we generate ourselves, ensuring they match the schema of the original tables while avoiding overlap with real Wikipedia content. This design choice limits the chance that the non-member data unintentionally contains information seen during pretraining.
>
> More generally, one of the strengths of Tab-MIA is that it provides a controlled, extensible framework that future work can use to explore alternative constructions for non-member data (e.g., DP-generated tables, schema-preserving perturbations) and deeper investigations of distribution-shift effects.
>
> **Q1:** Evidence that LLMs are trained on tabular data containing PII
>
> **A1:** We will strengthen the motivation by citing prior work showing that modern LLMs are trained on structured/tabular content that may contain PII. Prior research confirms that CommonCrawl-based pretraining includes large amounts of raw HTML with <table> elements and CSV-like structures, and that LLMs can memorize sensitive fields present in such data. In particular, Lukas et al. provide strong evidence of PII leakage from semi-structured datasets such as legal, medical, and email records [1]. Additional works show that LLMs routinely ingest serialized table formats—HTML, CSV, JSON—and learn from them: Sui et al. (2024) [2], Fang et al. (2024) [3], and Hegselmann et al. (2023) [4].
> We will incorporate these citations into the revised version to clarify the motivation.
>
> **Q2:** Add Ref/Ratio attack
>
> **A2:** We appreciate the suggestion. In our work we already evaluated a closely related reference-based method—the Zlib attack, as described in our paper. Zlib is conceptually similar to reference-based ratio-style attacks because it normalizes the model likelihood using compression-based estimates. However, in our experiments it consistently achieved lower performance compared to the simpler PPL (LOSS) attack.
>
> For example, on the WikiSQL dataset fine-tuned on Mistral 7B, using the Line-Separated encoding for 3 epochs:
>
> * Zlib achieved 67.8\% AUROC,
>
> * while PPL achieved 75.1% AUROC.
>
> Because Zlib underperformed across most datasets and models, we chose not to include its results in the main tables to preserve clarity. However, the full Zlib implementation is included in our released code, and researchers can easily run it using Tab-MIA.
>
> Bibliography:
>
> [1] Lukas, N., Salem, A., Sim, R., Tople, S., Wutschitz, L., \& Zanella-Béguelin, S. (2023). Analyzing Leakage of Personally Identifiable Information in Language Models. IEEE Symposium on Security and Privacy (SP).
>
> [2] Sui, Y., Zhou, M., Zhou, M., Han, S., \& Zhang, D. (2024). Table Meets LLM: Can Large Language Models Understand Structured Table Data? A Benchmark and Empirical Study. arXiv preprint arXiv:2305.13062.
>
> [3] Fang, X., Xu, W., Tan, F.A., Zhang, J., Hu, Z., Qi, Y., Sengamedu, S.H., \& Faloutsos, C. (2024). Large Language Models (LLMs) on Tabular Data: Prediction, Generation, and Understanding – A Survey. arXiv preprint arXiv:2402.17944.
>
> [4] Hegselmann, S., Buendia, A., Lang, H., Agrawal, M., Jiang, X., \& Sontag, D. (2023). TabLLM: Few-Shot Classification of Tabular Data with Large Language Models. Proceedings of AISTATS 2023.

---

> ### Comment · Reviewer_bFS1 · 2025-11-24
> **Response to author rebuttal**
>
> Thank you to the authors for their response. While several points have been addressed, some concerns/suggestions remain unresolved/unaddressed.
>
> (1) I am still unconvinced by the rationale for excluding model-less baselines from the results in Section 5.4, as used in prior work [1, 2]. It is well established that MIAs evaluated on data with distribution shift produce misleading and inflated results. When evaluated properly, MIA performance then collapses in many cases to a random-guess [1, 3]. The authors argue that *“To reduce this risk, we use synthetic non-member tables that we generate ourselves, ensuring they match the schema of the original tables while avoiding overlap with real Wikipedia content. This design choice limits the chance that the non-member data unintentionally contains information seen during pretraining.”* However, prior work has shown that synthetic data can also lead to such issues [4], and this argument alone does not justify not providing a model-less baseline. Importantly, such a baseline is extremely simple to implement, e.g. a random forest bag-of-words classifier as in [1, 2].
>
> (2) Further, although the zlib attack and the Ratio attack both normalize the target model loss by how 'normal' a piece of text is, they are still quite different. I strongly suspect that computing the Ratio attack (i.e., dividing the finetuned model’s loss by the pretrained model’s loss) would yield higher AUCs than any of the reported methods.
>
> I am pushing back on these two points because (1) is essential for interpreting the results in Section 5.4, and (2) would just strengthen the paper. Moreover, I believe that both experiments are quite inexpensive to implement.
>
> [1] Meeus, M., Shilov, I., Jain, S., Faysse, M., Rei, M., & de Montjoye, Y. A. (2025, April). Sok: Membership inference attacks on llms are rushing nowhere (and how to fix it). In 2025 IEEE Conference on Secure and Trustworthy Machine Learning (SaTML) (pp. 385-401). IEEE.
>
> [2] Das, D., Zhang, J., & Tramèr, F. (2025, May). Blind baselines beat membership inference attacks for foundation models. In 2025 IEEE Security and Privacy Workshops (SPW) (pp. 118-125). IEEE.
>
> [3] Maini, P., Jia, H., Papernot, N., & Dziedzic, A. (2024). LLM Dataset Inference: Did you train on my dataset?. Advances in Neural Information Processing Systems, 37, 124069-124092.
>
> [4] Naseh, A., & Mireshghallah, N. (2025). Synthetic data can mislead evaluations: Membership inference as machine text detection. arXiv preprint arXiv:2501.11786.

---

> > ### Author Response · Authors · 2025-11-24
> >
> > Thank you for the follow-up comments and for highlighting two important aspects of evaluating MIAs under distribution shift. We appreciate the opportunity to clarify our design decisions and to refine the paper accordingly.
> >
> > (1) Model-less baseline in Section 5.4
> >
> > We agree that blind/model-less baselines (e.g., bag-of-words + random forest, as in Meeus et al. 2025 and Das et al. 2025) are valuable for contextualizing results under potential distribution shift.
> > While our experiment in Section 5.4 attempts to mitigate this issue by constructing schema-preserving synthetic non-members, we fully acknowledge that synthetic data alone is not sufficient to eliminate distribution artifacts—consistent with findings such as Naseh & Mireshghallah (2025).
> >
> > To address this, in the revision we will:
> >
> > Add a blind, model-less baseline, following the setups of Meeus et al. (2025) and Das et al. (2025).
> >
> > Reinterpret Section 5.4 in light of this baseline, clearly distinguishing between
> > (a) true pretrained-model memorization signals, and
> > (b) signals potentially arising from distribution mismatches.
> >
> > This will improve the clarity and interpretability of the pretrained-model results.
> >
> > (2) Ratio attack (loss_FT / loss_PT)
> >
> > Thank you for emphasizing this point. Although we evaluated Zlib as a normalization-based baseline, the Ratio attack is conceptually different and can be computed efficiently using the pretrained checkpoints already present in our experiments.
> >
> > Including it will strengthen the benchmark and provide a more complete comparison.
> >
> > Accordingly, in the camera-ready version we will compute and include the Ratio baseline for all relevant experiments, reported in the appendix, and discuss its relationship to Zlib and PPL, and describe when Ratio provides a stronger signal.

---

### Official Review · Reviewer_oEMe · 2025-10-31

**Soundness:** 2
**Presentation:** 2
**Contribution:** 3
**Rating:** 2
**Confidence:** 4

**Summary:**

The paper introduces Tab-MIA, a benchmark dataset for evaluating membership inference attacks (MIAs) on large language models (LLMs) trained with tabular data. Using this benchmark, the authors conduct systematic evaluation of MIA methods across multiple tabular encoding/serialization formats. They find that LLMs can memorize structured data to an extent that enables effective membership inference, with AUROC scores approaching 90% after minimal fine-tuning. The study also reveals partial transferability of attacks across encoding formats, emphasizing the need for stronger privacy-preserving training practices when adapting LLMs to tabular data.

**Strengths:**

- The paper is easy to follow and presents a useful and timely contribution, introducing the benchmark for MIAs on tabular data in LLMs.

- Provides valuable empirical insights into memorization patterns and attack transferability, highlighting underexplored privacy risks in adapting tabular data to LLM.

- The methodology and benchmark design have practical utility for future privacy research on structured data.

**Weaknesses:**

- The bullet points in lines 257–269 repeat the content of Figure 1; consider moving these details to the appendix for brevity.

- Consider including a visual example contrasting long- vs short-context table encodings to help readers intuitively understand the setup.

- The paper relies heavily on MIA metrics but provides limited explanation of them. Expanding the description of attack metrics in the main text would make the results more interpretable.

- Consider reordering the results section, starting with Section 5.2 (encoding comparison) would align better with the paper’s central research question.

I am assigning a low score primarily due to concerns about reproducibility. For a benchmark paper, the release of code is essential to validate results and facilitate future research. Without access to the implementation, it is difficult to fully trust or reproduce the reported findings. I would be willing to raise my score if the authors make the code publicly available and provide clear implementation details of the MIA metrics in the paper, since much of the paper’s analysis depends on these metrics.

**Questions:**

- The filtering step (line 246) is unclear, why does the Adult dataset shrink from 48k to 3k samples? How exactly was the filtering applied to reduce the Adult dataset size?
- What would happen if models were fine-tuned using a combination of encoding formats, would this amplify or mitigate memorization?

---

> ### Author Response · Authors · 2025-11-15
> **Response to Reviewer**
>
> Thank you for the useful suggestions, particularly regarding clarity, reproducibility, and presentation. We address all points below.
>
> **W1:** Section 257–269 repeats Figure 1
>
> A1: Agreed. We will move these bullet points to the appendix to avoid redundancy.
>
> **W2:** Add a visual example contrasting long vs. short context
>
> **A2:** We will improve the clarification in the text by adding a concise visual example that contrasts short-context tables with long-context segmented tables. This will make the distinction more intuitive and help readers understand the preprocessing pipeline. Although the raw structures differ, both table types are ultimately converted into the same six serialized formats used in Tab-MIA, ensuring a uniform evaluation setup. This clarification will be added to the revised version.
>
> **W3:** Limited explanation of MIA metrics
>
> **A3:** We will expand Sec. 4 to better describe Min-K\%, Min-K++ normalization, and threshold-dependent metrics, as requested.
>
> **W4:** Recommend reordering results (start with encoding comparison)
>
> **A4:** We agree and will reorder Sec. 5 accordingly.
>
> **W5:** Reproducibility concerns; code not visible
>
> **A5:** The full code is already publicly available at our anonymous repository:
> https://anonymous.4open.science/r/Tab-MIA
> We did not include a link in the submitted PDF due to the anonymity and formatting constraints required by the submission instructions. In the camera-ready version, we will add a link to the public GitHub repository and include detailed reproduction commands, scripts, and environment specifications directly in the appendix to ensure full reproducibility.
>
> **Q1:** Adult dataset shrinks from 48k to 3k—unclear
>
> **A1:** The reduction comes from the segmentation step applied to long-context tables. The Adult dataset is originally a single table with 48k rows, which cannot fit into the model’s context window when serialized. To ensure consistent and tractable inputs, we divide it into 20-row segments, resulting in approximately 2,440 unique table chunks that serve as member and non-member samples. We will clarify this preprocessing step more explicitly in Sec. 3.2.
>
> **Q2:** What if fine-tuned using multiple encodings?
>
> **A2:** Our hypothesis is that each table would still be memorized according to the specific encoding in which it appears, because memorization is tightly linked to tokenization patterns and the structural regularities introduced by each format. Moreover, training the same table in different encodings would likely degrade memorization, since the model is exposed to multiple, non-aligned structural representations. This scenario resembles the cross-format mismatch studied in Section 5.3, where evaluation in a different encoding than the training encoding leads to lower attack performance. Similarly, mixing encodings during fine-tuning would make it harder for the model to learn a consistent ordering or representation of the table, thereby reducing direct memorization.

---

### Official Review · Reviewer_P5qU · 2025-10-31

**Soundness:** 3
**Presentation:** 3
**Contribution:** 2
**Rating:** 4
**Confidence:** 4

**Summary:**

This paper introduces Tab-MIA- a dataset for evaluating membership inference of tabular data in LLMs. Tab-MIA consists of a set of tables sourced from public datasets, represented in a variety of common formats used for LLM training. The authors present an overview of the dataset and run a series of experiments to demonstrate its utility for auditing publicly trained and privately fine-tuned models.

**Strengths:**

* The work fills an important gap in auditing LLMs for privacy risks, beyond conventional text membership tests.
* Strong evaluation on a suite of LLMs under a fine-tuning paradigm, as well as evaluation of some public pretrained models.

**Weaknesses:**

* I wonder about the realism of fine-tuning a model on some context-free tables- in most natural language tasks a table will be associated with accompanying text to set context, provide explanation, inject a user query, etc.  From this perspective I'm not sure how to interpret the significance of the attack results.

* Experiments on larger models would be valuable, even just targeting large pretrained models without fine-tuning.

* An experiment demonstrating a defense (eg training with DP-SGD or privatizing the data itself with a DP method) and potential privacy-utility tradeoffs would strengthen the paper.

**Questions:**

1. How do you work around limitations of some formats- eg lack of ability for some formats to allow spans across multiple columns or embed sub-tables?  The universe of tables in public data, especially html-formatted tables, is much richer than simple grid format.

2. Aside from table representation do you have any other insights as to the kinds of tables that are easily memorized?

3. Generating synthetic non-member tables from Gpt-4o-mini: these are clearly contaminated by both the in-context sample and any public tables in GPT-4o's training set. I wonder If there are alternatives that would generate high-quality, plausble tables that aren't influenced by the in-member data.  One option that comes to mind is using a library like SmartNoise to generate tables from public seeds that meet a row-wise differential privacy guarantee, but would still be some level of contamination.

4. Evaluation against larger models and/or a public model API that provides logits for a very large model would be nice to have.

---

> ### Author Response · Authors · 2025-11-15
> **Response to Reviewer**
>
> Thank you for the thoughtful review and for highlighting directions for strengthening realism and extensions. We respond to each point below.
>
> **W1:** “Context-free table fine-tuning may be unrealistic”
>
> **A1:** We intentionally remove all textual context to precisely isolate the memorization of tabular structure and cell-level values, without interference from natural-language artifacts. This controlled setup is consistent with prior MIA benchmarks (e.g., WikiMIA and BookMIA isolate text segments without prompts) and is essential for understanding how LLMs memorize tables themselves rather than surrounding instructions. One of the strengths of Tab-MIA is that it provides a clean, decoupled environment where the effects of encoding choices, row uniqueness, and table structure can be analyzed independently—something that cannot be done when natural language context is mixed in.
> Realistic, mixed table–text scenarios (e.g., instructions or questions paired with tables) are indeed relevant, and our benchmark is designed so these settings can be added as extensions in future work.
>
> **W2:** Larger models would be valuable
>
> **A2:** We agree that evaluating larger models is important. Our current benchmark focuses on finetunable open-weight models. However, Tab-MIA already supports attacks on pretrained models of any size, including API-only models, as long as token probabilities or log-likelihood estimates are accessible.
> In fact, one of the advantages of our dataset is that it is format-agnostic, lightweight, and modular, making it easy to evaluate leakage on larger models without fine-tuning (as we already demonstrated in Sec. 5.4).
>
> **W3:** Lack of DP or other defenses in experiments
>
> **A3:** Evaluating defenses such as DP-SGD is an important direction, and our benchmark is specifically designed to support such analyses. While the current paper focuses on establishing the benchmark and demonstrating its ability to compare MIA methods across formats and models, we view the systematic evaluation of privacy defenses—including DP-SGD and data-level anonymization techniques—as natural future work. The infrastructure introduced in Tab-MIA already provides the foundation needed to measure the privacy–utility trade-offs of different defenses. In the revised version, we will explicitly discuss how Tab-MIA enables these studies and outline defense evaluation as a key extension of the benchmark.
>
> **Q1:** Limitations of formats (no spans/subtables)?
>
> **A1:** The benchmark uses flat tables because all five source datasets are flat. Multi-span and nested-table support will be included in a future release; we will clarify this limitation.
>
> **Q2:** Which tables are more easily memorized?
>
> **A2:** Our results indicate that row uniqueness, row length, and the density of informative tokens strongly influence memorization. Tables containing many unique or high-entropy rows provide more opportunities for the model to memorize specific patterns, leading to higher AUROC values. Conversely, tables with repetitive or low-information rows tend to exhibit lower leakage. We will add a brief analysis highlighting these correlations and include supporting visualizations (e.g., correlation plots) to clarify which structural properties make certain tables more susceptible to memorization.
>
> **Q3:** Synthetic non-members may be contaminated by GPT-4o-mini’s training data
>
> **A3:** We acknowledge this concern. The synthetic non-member tables are used only in the pretrained-model setting, where we cannot control the model’s original training corpus and therefore need a clean separation between member and non-member distributions.
> Importantly, generating high-quality synthetic non-members using a separately trained model is non-trivial for short structured tables: their heterogeneous schemas, small row counts, and diverse attribute combinations make it difficult to train a standalone generative model that produces realistic but distributionally matched samples. GPT-based generation was chosen because it reliably preserves the schema and structure while avoiding direct overlap with naturally occurring Wikipedia tables. We will clarify this design choice in the revised version.
>
> **Q4:** Evaluation against larger models or API logits
>
> **A4:** We agree that evaluating larger models is important. At present, however, most large API-based models do not provide logit access, which makes it impossible to run standard reference-free MIAs such as LOSS or Min-K\% on them. Our benchmark is designed so that, once such access becomes available, evaluating larger models (e.g., 30B–70B Llama variants) will be straightforward.

---

### Official Review · Reviewer_petP · 2025-11-04

**Soundness:** 2
**Presentation:** 2
**Contribution:** 2
**Rating:** 0
**Confidence:** 4

**Summary:**

This paper introduces a benchmark dataset, Tab-MIA, for evaluating MIAs on tabular data in LLMs. Tab-MIA includes short-context and long-context tables, each represented in six different textual encoding formats.


Short-context tables are derived from QA benchmarks WikiTableQuestions (WTQ), WikiSQL, and TabFact. Long-context tables are derived from structured tabular benchmarks frequently used in fairness, regression, and privacy studies: Adult (Census Income) dataset and the California Housing dataset.

Encoding fromats are: JSON, HTML, Markdown, Key-Value Pair, Key-is-Value, and Line-Separated (CSV-like).

**Strengths:**

- Running three exisitng MIAs (LOSS attack, Min-K% attack, Min-K%++ ) on LLaMA-3.1 8B, LLaMA-3.2 3B, Gemma-3 4B, and Mistral 7B QLoRA fine-tuned on Tab-MIA dataset.

- Highlighting that Tabular data may contain personally identifiable information (PII), commercially sensitive material, or domainspecific details that are not intended for broad dissemination

**Weaknesses:**

- Lack Dataset novelty/validity: Tab-MIA is a recombination of existing tabular datasets. The construction also risks the member vs. non-member boundary, as there are no guarantees that chosen LLMs have not already seen all data.

- No methodological novelty: No new membership-inference attack tailored to tabular data is proposed.

- Results lack novelty: The paper largely reiterates established findings, nothing new about tabular data:
	- LLMs can memorize tabular data
	- partial transferability of attacks across encoding formats
	- a consistent and substantial increase in vulnerability as the number of fine-tuning epochs grows
	- larger models are also significantly more vulnerable to MIAs

- Use of pre-training and fine-tuning is not consistent

- Typo in line 121: DC-PDD(

**Questions:**

See above

---

> ### Author Response · Authors · 2025-11-15
> **Response to Reviewer**
>
> Thank you for your review and for highlighting concerns regarding novelty, methodology, and result interpretation. We address each point below, and we kindly ask you to reconsider your evaluation, as the benchmark introduces meaningful novelty and plays an important role in addressing the growing use of LLMs with tabular data.
>
> **W1:** Dataset lacks novelty; recombination of existing datasets; Risk that LLMs have already seen the data
>
> **A1:** Our goal is not to create new tabular content but to provide the first controlled benchmark enabling systematic, cross-format evaluation of MIAs on tabular data. Existing MIA benchmarks (WikiMIA, BookMIA, MIMIR) contain no tabular data and do not provide multi-format serializations, deduplication pipelines, or rigorously paired member/non-member splits. In Tab-MIA, member samples are explicitly defined as the tables used to fine-tune the model, while non-member samples are taken from the held-out portion of the dataset—ensuring a clean and reproducible setup essential for MIA evaluation. This controlled construction is not available in prior benchmarks.
> For the pretrained-model experiments, we additionally use synthetic non-member tables generated by GPT-4o-mini. This avoids accidental overlap with the model’s pretraining corpus and guarantees that the non-member set does not contain naturally occurring Wikipedia tables. This design enables a fair comparison between member and non-member distributions even when we cannot control pretraining data.
>
> **W2:** No methodological novelty (no new MIA tailored to tabular data)
>
> **A2:** The submission is to the Datasets and Benchmarks track. The novelty lies in creating the first benchmark for structured tabular data, enabling reproducible MIA analysis across formats, cross-format transfer studies, and pretrained-model leakage evaluation.
>
> **W3:** Results lack novelty; trends are already known
>
> **A3:** While some high-level trends (e.g., model size and epoch count) are known for text, no prior work has studied these effects on tabular data, nor the new phenomena we reveal:
> * encoding-sensitive memorization differences,
> * cross-format transferability,
> * pretrained leakage on tabular Wikipedia-derived tables.
> These results have not been previously reported.
>
> **W4:** Use of pretraining and fine-tuning not consistent
>
> **A4:** They address different research questions:
>
> Fine-tuning experiments measure induced memorization.
> Pretrained experiments measure existing memorization of public tabular corpora.
> We will clarify this distinction.
>
> **W5:** Typo (DC-PDD)
>
> **A5:** Corrected—thank you.

---

> > ### Comment · Reviewer_petP · 2025-11-25
> > **Response to Author rebuttal**
> >
> > Unfortunately, my concerns have not been addressed, especially these two:
> >
> > > **W4: Use of pretraining and fine-tuning not consistent. A4: They address different research questions: Fine-tuning experiments measure induced memorization. Pretrained experiments measure existing memorization of public tabular corpora. We will clarify this distinction.**
> >
> > I think the authors misunderstood my comment, for example, see one confusing bit with broken flow in the abstract where you talk about fine-tuning in the first sentence and then talk about pre-training in the next sentence:
> >
> > "Using our Tab-MIA benchmark, we conduct the first evaluation of state-of-the-art MIA methods on LLMs finetuned with tabular data across multiple encoding formats. In the evaluation, we analyze the memorization behavior of pretrained LLMs on structured data derived from Wikipedia tables."
> >
> > > **In Tab-MIA, member samples are explicitly defined as the tables used to fine-tune the model, while non-member samples are taken from the held-out portion of the dataset**
> >
> > This clarification actually highlights my core concern rather than resolving it: how do you make sure that this held-out portion of the dataset has not already been seen by your chosen LLM or even worse, by other people's LLMs who are interested in using your dataset.
> >
> > So I keep my score.

---

> > > ### Author Response · Authors · 2025-11-25
> > >
> > > Thank you for pointing this out, and we apologize for not fully addressing your original concern. Your clarification helped us understand that the core issue is not the distinction between fine-tuning and pretraining experiments themselves, but the way the manuscript presents them and the potential confusion this causes for readers.
> > >
> > > First, regarding the inconsistent flow in the abstract: we agree that the current phrasing mixes fine-tuning and pretrained-model results too closely. In the revised version we will clearly separate these two evaluations so that the narrative is consistent. Specifically, we will describe the fine-tuning experiments first, which constitute the main controlled MIA setting in Tab-MIA, and then explicitly introduce the pretrained-model analysis as a complementary, secondary experiment with different assumptions and goals.
> > >
> > > Second, you are correct that our explanation of member and non-member splits primarily clarifies the fine-tuning setting and does not directly address the concern of possible pretraining contamination. While it is theoretically possible that a pretrained LLM may have encountered parts of a public dataset before, the fine-tuning experiments in Tab-MIA do not rely on the pretrained model’s prior exposure. In our setup, the model is fine-tuned for several epochs on a clearly defined member subset, which causes the model to shift its distribution and focus strongly on this new training data. Prior work has shown that repeated gradient updates during fine-tuning significantly amplify memorization of the fine-tuning data, even when starting from a pretrained model.
> > > In other words, even if some rows existed in pretraining, the fine-tuning process overwrites the model’s attention toward the newly provided member samples, letting us cleanly measure induced memorization.

---

### Meta-Review · Area_Chair_zTyz · 2025-12-27

**Summary:**

This paper introduces a new MIA benchmark on tabular data, addressing a gap in the literature given that many sensitive datasets are naturally structured. In particular, the paper studies how different table encodings (e.g., JSON, Markdown, HTML, line-separated formats) affect MIU vulnerabilities – an issue largely unexplored in prior work. The benchmark spans multiple domains (Wikipedia, US Census, US Housing Survey) and evaluates well-established MIA methods, including PPL, Min-K, and Min-K++. Results show that (1) MIA accuracy can be as high as 90% (notably for Mistral), highlighting substantial MIA risks, and (2) vulnerability varies significantly by encoding formats, with Markdown, key-value, and line-separated formats generally yielding high attack access. The paper also includes ablations on known factors such as the number of epochs.

Reviewers found that the idea of specifically looking at privacy risks in tabular/structured data is novel and interesting (bFS1, P5qU, petP), the paper is well-written (bFS1, oEMe), related work is comprehensive (bFS1), experiments are thorough (bFS1, petP), and the benchmark and analysis are practically valuable (oEMe).

Concerns that are resolved
- Lack of reproducibility (oEMe): The authors provided code and dataset as supplementary materials.
- Realism of fine-tuning a model on context-free tables (P5qU): In AC’s opinion, authors’ responses make sense, and prior MIA benchmarks similarly operate on raw data rather than tasks.
- Lack of experiments on larger models (P5qU): The paper includes up to 7B models, and it is straightforward to apply it to larger models. Larger models will likely to strengthen, rather than overturn, the paper’s conclusions on MIA risks on tabular data.
- Missing experiments with defenses such as DP (P5qU): While a valid concern, incorporating DP is likely a non-trivial future work, and prior MIA studies mostly omit it either.
- Lack of novelty in results (petP): Author responses clarify that results include novel findings such as impact of encoding, cross-format transferability, and more.
- Lack of novelty in methods, as it includes existing MIA methods (petP) - As this is a Dataset and Benchmark track, proposing new MIA methods is not required.

Concerns that may or may not have been resolved, mainly because authors acknowledged the weaknesses and promised revisions, but the manuscript wasn’t actually updated.
- Too many portions of discussion reiterate known MIA trends, such as the effect of training epochs, and are less specific to tabular data (bFS1). The AC agrees with the reviewer comment and recommends reducing this content.
- Differences in MIA risks across datasets and encoding formats are more important, but under-discussed. For instance, discussion on different datasets is missing; the intuition on why certain formats are more risky than others is not very convincing (bFS1) – The AC agrees with the reviewer comment and recommends substantially expanding the relevant discussion.
- Results in Section 5.4 are questionable, and model-less baselines should be included (bFS1)

Concerns that are unlikely to have been resolved (in AC’s opinion)
- The dataset novelty is somewhat limited given that it is a combination of existing datasets, and the paper did not make sure members and non-members boundaries are clear, as it could have been contaminated (petP): The AC agrees with this comment. Although the main target is a fine-tuning setup, private data would normally be unseen during pre-training. Using Wikipedia tables is fine as it is a reasonable source of tabular data, but the paper could have taken stronger steps to ensure non-members are true non-members – for instance, by restricting non-members to be those posted after the known pre-training cutoff, as done in WikiMIA/BookMIA.

**Reviewer Concerns:**

See above

**Reviewer Scores:**

See above

---

### Decision · Program_Chairs · 2026-01-26

Accept (Poster)